# Neurobehavioral Alterations from Noise Exposure in Animals: A Systematic Review

**DOI:** 10.3390/ijerph20010591

**Published:** 2022-12-29

**Authors:** Giulio Arcangeli, Lucrezia Ginevra Lulli, Veronica Traversini, Simone De Sio, Emanuele Cannizzaro, Raymond Paul Galea, Nicola Mucci

**Affiliations:** 1Department of Experimental and Clinical Medicine, University of Florence, 50139 Florence, Italy; 2Occupational Medicine School, University of Florence, 50139 Florence, Italy; 3R.U. of Occupational Medicine, “Sapienza” University of Rome, 00100 Rome, Italy; 4Department ProMISE, University of Palermo, 90100 Palermo, Italy; 5Department of Obstetrics & Gynaecology, University of Malta, MSD 2080 Msida, Malta

**Keywords:** noise, environment, fauna, animals, neurobehavioral disorders, exposure, acoustic pollution, prevention, risk

## Abstract

Ecosystems are increasingly involved and influenced by human activities, which are ever-increasing. These activities are mainly due to vehicular, air and sea transportation, thus causing possible repercussions on the fauna that exists there. The aim of this systematic review is to investigate the possible consequences that these activities may have in the field of animal neurobehavior, with special emphasis on the species involved, the most common environment concerned, the noise source and the disturbance that is caused. This research includes articles published in the major databases (PubMed, Cochrane Library, Scopus, Embase, Web of Sciences); the online search yielded 1901 references. After selection, 49 articles (14 reviews and 35 original articles) were finally scrutinized. The main problems that were reported were in relation to movement, reproduction, offspring care and foraging. In live experiments carried out, the repercussions on the marine environment mainly concerned altered swimming, shallower descents, less foraging and an escape reaction for fear of cetaceans and fish. In birds, alterations in foraging, vocalizations and nests were noted; laboratory studies, on the other hand, carried out on small mammals, highlighted spatio-temporal cognitive alterations and memory loss. In conclusion, it appears that greater attention to all ecosystems should be given as soon as possible so as to try to achieve a balance between human activity and the well-being of terrestrial fauna.

## 1. Introduction

The impact of human-caused environmental pollution affects nature and other living species through difficulties in the food supply, behavioral changes in predation, mating and migratory phenomena. Noise pollution is a growing concern in public and environmental health. It can be defined as the emission in the environment of any source of anthropogenic sound that can have detrimental effects on the health and comfort of people and animals, natural resources, and the ecological balance of an area [1]. Since the Industrial Revolution, the level of noise both in terrestrial and marine environments has dramatically increased. For example, researchers estimate that oceans’ levels of sound are 2 to 10 times higher than before the beginning of industrial activities [2,3]. Noise pollution expands with human population growth, globalization of transportation networks, expansion of resource extractions, and urban and industrial development [4,5]. This is responsible for chronic noise exposure in most terrestrial areas, including remote wilderness sites, which threaten ecological integrity and contribute to climate change in habitat destruction [6]. Models show that no area of oceans is unaffected by human influence and that a large fraction (41%) is strongly affected by multiple drivers [7].

Laboratory studies and field research have identified four main ways in which animals are adversely affected by noise pollution: (i) hearing loss, with noise levels of 85 Decibel or higher; (ii) masking, such as the inability to hear important environmental and animal signals; (iii) increased heart rate and breathing; and (iv) behavioral effects. There is considerable intra-species variability, and this varies according to the characteristics of the noise. This may lead to territory abandonment and loss of reproduction [8].

In recent years, one particular scenario that has been put to the test is the marine environment. Marine life is threatened by habitat degradation due to human activities such as fishing, ship traffic, pollution, coastal anthropization, and high noise levels due to propellers and diesel engines [9].

Marine animals such as whales, which depend on sound for communication, can be affected by noise in various ways. Marine mammals live in a habitat that transmits little light but through which sound propagates well and quickly, even over great distances. For this reason, marine mammals rely on sound to communicate, explore the environment, find their prey and avoid obstacles. Research had shown that higher ambient noise levels also caused the animals to vocalize louder (“Lombard Effect”) and that the duration of the humpback whales’ song was longer when there was low-frequency sonar in the vicinity [10].

Exposure to noise can produce a wide range of effects on marine mammals. The low-level sound may be audible to animals without producing any visible effects; higher-intensity sound can disturb the animals, causing them either to move away or may produce other behavioral changes. Noise can increase the risk of death by modifying the delicate balance between predators and prey, interfering with the use of sounds in communication, especially in relation to reproduction and navigation [9].

Hearing overexposure can also lead to temporary or permanent hearing loss. It has been shown that European robins living in urban environments are more likely to sing at night when they are in locations with high levels of daytime noise pollution. This is due to the fact that at night their message can carry through the environment more clearly. The same study showed that daytime noise was a stronger predictor of night singing than, for example, nighttime light pollution [11]. Other behavioral changes involve reproduction. Some studies have shown that zebra finches become less loyal to their partners when exposed to vehicular traffic noise. This could alter the evolution of a population by selecting some genetic traits over others, weakening the resources normally dedicated to other activities and thus leading to profound evolutionary genetic consequences [12].

Noise pollution contributes to detrimental effects both on humans and animals, as are many other emerging environmental threats, such as climate change or hydrogeological instability. Human, animal and ecosystem health are inextricably linked; therefore, a comprehensive approach, such as that provided by the “One Health” holistic vision is fundamental in a globalized world [13]. A previous review by the same authors addressed the neurobehavioral effects of professional noise on humans [14]. To complete the framework of detrimental effects of noise pollution on living beings, this systematic review aims at analyzing what the main neurobehavioral disorders in animals exposed to noisy anthropogenic sources are, analyzing the most recent scientific literature. For this purpose, basic descriptions of underlying biological mechanisms will be provided when needed for a more comprehensive understanding of the phenomenon without the claim of delivering an in-depth analysis from a pathophysiological point of view.

## 2. Materials and Methods

This review of the scientific literature is compliant with the PRISMA guidelines (Preferred Reporting Items for Systematic Reviews and Meta-Analyses) [15].

### 2.1. Literary Research

The search included articles published in the last 12 years, from 2010 to July 21st 2022, on the main online databases (Pubmed, Scopus, Embase, Web of Science, Cochrane Library). In addition, a manual search of the selected articles and reviews was carried out to check for any other eligible paper that the online research might have missed (0). The literature search was restricted to the last 12 years so as to depict an updated state of the art. The literature search used a series of keywords, alone and in combination with each other: noise, loud, sound, exposure, environment, neurobehavioral, psychological, mental, and neural. Appendix A includes the specific string used for each searched database. 

The PICO scheme shown in Table 1 was used on the data to study the effects of noise exposure on animals. The authors included all animals without considering any difference in species, habitat or geolocation. Intervention is the exposure to any type of noise (e.g., noise pollution caused by human vehicles, exposure to noise in experimental settings) in any terrestrial environment. The health outcomes that were considered were those related to neurobehavioral changes, as reported in recent studies. In particular, the authors included short-term effects and medium- to long-term effects, such as changes in swimming direction and speed, burrow building, foraging for food and reproduction, irritability, agitation, anger, and changes in neurobehavioral skills. If available, hormone excretion levels or alterations in instrumental diagnostic tests were also included. The PICO scheme that was followed is summarized in Table 2. 

Two independent reviewers read the titles and the abstracts of the various studies identified by the database search. Inclusion and exclusion criteria pertinent to this study were used by the reviewers to select the studies that were used. The opinion of a third researcher was sought when discordancy between the first two researchers existed. Subsequently, the authors independently reviewed the complete texts of the selected studies so as to decide on final admissibility. Finally, the authors eliminated duplicate studies and articles whose full text was not available. Relevant data were collated on a spreadsheet, and this included the date and country of publication, the animal species examined, the noise level (if available), and the type of disorders reported.

### 2.2. Quality Assessment

Two different reviewers assessed the methodological quality of the selected studies using specific assessment tools so as to reduce the risk of introducing any bias. The opinion of a third reviewer was sought when divergence of opinion existed between the first two reviewers. The INSA “International Narrative Systematic Assessment” method was used to judge the quality of narrative reviews [16], the AMSTAR (A Measurement Tool to Assess systematic Reviews) to evaluate systematic reviews [17] and the Newcastle Ottawa Scale to evaluate cohort studies and control case studies [18]. The JADAD scale was applied for randomized clinical trials [19].

### 2.3. Eligibility and Inclusion Criteria

In this review, the studies that were included focus on noise exposure and the animal species exposed to this risk. Studies on the main neurobehavioral consequences of this exposure, in particular aggression, adaptation systems, nutrition, reproduction and anti-predatory behaviors, were included. All types of study designs were also included. Only articles written in English were included. 

### 2.4. Exclusion Criteria

Publications concerning just human subjects were excluded. Furthermore, publications that did not report neurobehavioral alterations were excluded, as were editorials. Studies of little academic relevance and individual contributions were also excluded. Descriptive studies presented at scientific meetings not having any quantitative or qualitative scientific elements were also deemed to be ineligible. Papers published prior to 2010 were excluded from this study so as to be able to focus on the more recent evidence available in the literature, thus allowing a more effective synthesis of the current knowledge in the field. 

## 3. Results

The online search yielded 1901 studies: PubMed (469), Scopus (712), Embase (81), Web of Science (360), and Cochrane Library (9). Forty-nine studies were included in this literature review after eliminating duplicates (806). Seven-hundred-and-fifty-seven (757) deemed to be unrelated to alterations associated with noise exposure in animals, as assessed through the title and abstracts of the articles (757), were also excluded. Furthermore, the full article was not available in six (6) cases, and fourteen (14) did not meet the inclusion/exclusion criteria. The selection process is graphically represented in Figure 1. 

There were 34 original articles, twelve (12) narrative reviews, two (2) systematic reviews and one (1) meta-analysis in the articles reviewed. Of the original articles, fourteen (1) were experimental studies, nine (9) were observational studies, another nine (9) were case-control studies and two (2) were cohort studies (Table 2).

Twelve (12) articles, or 24.4% of the articles, were published in the United States. Most of the articles were published in 2020 (11 studies; 22.4%), followed by 2021 (10 articles; 20.4%).

The main neurobehavioral disorders present in the groups of animals examined were alterations in movement or swimming, with changes in nutrition, reproduction or anti-predator adaptations (32 articles; 65.3%), aggression, hyperactivity and overt anxiety (9 articles; 18.3%) and, alteration of autonomic reflexes and memory abilities (8 articles; 16.3%).

Marine fauna constituted the main animal group that was examined. This group included fish, whales, dolphins and cephalopods) where no less than twenty-five (25 or 51%) articles were retrieved. Eleven (11 or 22.4%) studies concerned the next group made up of rats, mice and rodents. Ten (10 or 20.4%) articles dealt with other mammals of various sizes (such as dogs, zebras, elephants, and cows) and four (4 or 8.1%) articles dealt with birds.

The main results arising out of the reviews of the scientific papers analyzed are described below and summarized in Table 3 and Table 4. Table 4, in particular, summarizes the original articles and provides information about the sample, noise level and study length. The study length provides information about the timing of the effects described. 

### 3.1. Reviews

The reviews included in this paper show how exposure of fauna to noise can have negative repercussions. These repercussions vary in severity, ranging from slight physiological alterations to extremely stressful conditions.

Di Franco highlighted how noise could have an impact on the physiology of various invertebrates (decapod crustaceans, cephalopods and cnidarian mollusks); this can range from an increase in stress-related variables to permanent structural damage, with possible fatal effects. In particular, crustaceans and cephalopods have manifested varying degrees of harm, from changes in movement to increased latency in response against predators. This can have an impact and alter the species’ reproduction and survival [5]. The effects of noise on impaired speech behavior, reduced numbers of species in noisy habitats, and changes in alertness and foraging behavior were also reported by Shannon et al. in their review spanning two decades. They found that anomalous responses from wildlife begin at noise intensities of around 40 dBA [20].

The environments that are most sensitive to anthropogenic noise are the marine ones. Kunc et al. have described how noise can negatively influence the perception of stimuli and the sense of orientation, causing possible strandings in both invertebrates and vertebrates. Cuttlefish, for example, change their visual cues when exposed to noise, and aquatic mammals can alter the use of their communication channels. On the other hand, in dolphins, noise decreases the accuracy of object detection. Noise pollution can also alter the avoidance of the noise itself, with possible negative consequences for the defense mechanisms against egg predators, for the maintenance of the territory, the choice of the mate and the care of the offspring [21]. In the reproductive field, malformations in marine invertebrate larvae during development have also been reported [22].

Anthropogenic noise can also compromise other important features. It can change swimming depth, directional changes, schooling adjustments and swimming speed [23]. Nabi et al. found that marine mammals such as whales alter their behavior, increasing the duration and speed of the dive and altering respiratory synchrony. A 50% reduction in foraging efficiency in whales has also been reported due to ship noise. This may result in decreased or even no nutritional intake and a consequent decrease in energy production [24]. Peng et al. found that boat disturbance reduced nest digging, defensive behavior against predators on eggs, and increased aggression. However, Peng et al. also pointed out that marine species can practice coping strategies. These include modifications of the sounds emitted or that the same reactions can depend on many variables such as position, temperature, physiological state, age, size, and distance (a strong behavior of avoidance would only be expected within 20 km of the noise source) [25].

Similarly, Popper et al. highlighted the same changes but highlighted that with acute noises in the first minute of exposure, there is a significant reduction in the swimming speed of the fish, while with continuous noise, greater alterations in foraging and in avoidance of predators are seen, as they are unable to perceive the sounds emitted by the predictors during migrations [26]. With regards to routes, Li et al. have described studies in which dolphins changed speed and orientation in response to the presence of small pleasure boats and fishing boats, and this up to a 1 km radius range [27]. 

Marine animals such as cuttlefishes can also modify their behavior, and Samson et al.l have reported studies showing that the hair cells in the statocysts and epidermal lines of S. officinalis and other cephalopods which could be the basis for directional hearing and sound localization in these animals are polarized. Cuttlefishes show alterations in inking and swimming speed as well as changing the swimming direction upwards, possibly to benefit from the sound shadow near the surface of the water. Variations in the respiratory rate during exposure to sound between 50–283 Hz [28] were also noted.

Kight et al., in their paper, describe how rats were trained to use visual cues in order to locate a submerged platform in a swimming pool. Animals exposed to conditions of loud noise during the learning phase of the experiment took longer to find the platform and spent less time in the target quadrant. Similarly, offspring of noise-stressed rats were also found to perform worse on visuospatial tests, with higher error rates [29].

However, not all noise emissions have a negative effect on animals; thus, in the article reported by Mandel et al., cows exposed to classical music (during the milking period, for a period of 28 weeks) had a higher milk production rate than those not exposed to such sound (6.27 min vs. 6.68 min, respectively) [30].

### 3.2. Experimental Studies

The exposure of rats to noise, in the experiment by Akefe et al., induced significant neurobehavioral deficits and the appearance of oxidative stress; administration of kaempferol and zinc gluconate reduced these noise-induced alterations, increasing the functionality of certain enzymes such as GPx, catalase and SOD, and reducing NO and MDA levels (*p* < 0.05 and <0.01, respectively). Administration of the active ingredients significantly improved the rats’ performance in the open field, motor coordination, motor strength, sensory-motor reflexes, and learning and memory (*p* < 0.05) [31]. Starting from the hypothesis that intense noise may cause hyperactivity of central auditory neurons by NMDA receptors and synaptic alterations, Criddle et al. actually noted how, by early administration of such receptor antagonists, animals showed fewer signs of agitation despite exposure to noise sources between 25–45 dB [32]. Furthermore, in the study by Uran et al., the neurological alterations in rats exposed to noise were examined. The noise frequencies between about 95–97 dB for 2 h or between the 15th and 30th postnatal days were investigated. Alterations in the different hippocampal regions were highlighted, together with significant behavioral abnormalities [33].

Other areas of the nervous system are also probably involved; for example, the locus coeruleus is an important pontine nucleus modulator of sympathetic tone, alertness and attention. There is a lot of evidence showing that hyperactivation of these areas, with increased release of norepinephrine in the locus coeruleus, causes fear/anxiety in experimental animals exposed to stressful stimuli. Koorpivaara et al. showed the anxiolytic effect of dexmedetomidine on dogs when compared to dogs given a placebo (OR3.5, CI 1.84–5.74, *p* < 0.0001). The Beagle dogs treated with dexmedetomidine exhibited less panting (*p* < 0.0001), less tremor (*p* = 0.0056), less vocalization (*p* = 0.0084) and inappropriate urination (*p* = 0.0314) compared to those who were administered a placebo [34].

Other researchers considered other hormonal alterations. Thus Mills et al. investigated the effects of speedboat noise on reproduction, behavior, cortisol and androgen levels in certain groups of anemones. The anemone fish exposed briefly to the noise generated by speed boats showed increased concealment and aggression signs. No effect on cortisol levels was noted, but male androgen levels (11-ketotestosterone and testosterone) were increased. In the long term, this led to higher cortisol levels in both sexes and higher testosterone levels in males [35]. In the study reported by Mikolajczak et al., 40 domestic geese were divided into two equal groups: the first group remained within 50 m of a wind turbine, and the second group was placed within a distance of 500 m. During the 12 weeks of the study, weight gain and blood cortisol concentration were assessed. The geese in the first group, that is, those most exposed to noise and vibrations, gained less weight and had a higher concentration of cortisol in the blood compared to the individuals in the other group [36].

Noise can also interfere with basic activities, such as movements, reproduction and foraging. Blatzer et al. highlighted how some groups of sole have significantly increased swimming speed when exposed to anthropogenic noises, such as occur in ports and construction sites within a radius of 132 to 766 m; similarly, cod showed a similar reaction, but the results here were not significant. The reactions depend on the context, age, water temperature, location and physiological state [37]. Likewise, some researchers examined the overall effects that exposures to sound have on swimming patterns in groups of sea bass. The fish increased their swimming depth; thus, they swam deeper but without increasing speed or losing cohesion with the group, particularly in the first few minutes of exposure, as if there was an initial fear reaction [38,39].

Noise can also alter the relationship between specimens. Frouin-Mouy et al. examined behavioral alterations in grey whales subjected to a UAV acoustic signal (Unmanned Aerial Vehicle provided by drone technology), and they found that a croak signal was produced just before the mother whale undertook a bubble blast beneath the calf. These underwater bubble blasts are estimated to have a detection range of 378 ± 134 m, given a background noise level of 96 ± 2 dB [40]. Female whales can reduce nestling provisioning rates during noise treatment (β ± SE = −0.066 ± 0.023, *p* = 0.018), and these whales were found to be slower in returning to the nest, particularly in the first year of exposure [41]. Female whales and pups were found to spend more time awake (+13.8–26.3%), more vigilant (+7.6–31.2%), spent more time in locomotion (+2.1–4.6%), with a reduction in nursing time (5.9–19.8% decrease associated with car noise and 15.4–31.8% decrease when noise was generated by boats) [42]. 

Leduc et al. noted that exposures to the additional noise did not yield any observable impairment of cognitive performances required to solve a spatial task. There was no difference in time required to achieve specific performance (reaching a target area in a T-maze) between noise-treated fish (100 dBA) and fish in control conditions (45 dBA) [43]. On the other hand, Issad et al. demonstrated that anxiety, such as behavior, and a significant decrease in activity in the wild desert rodent Gerbillus tarabuli might appear possibly due disturbance of components of the circadian rhythm [44].

### 3.3. Other Articles

Some researchers have focused on the effects caused by the interference of anthropogenic noise on animals, finding partially conflicting results. 

Abdullah et al. showed that elephants exposed to noises such as axes, vehicles, and chainsaws between 20–70 dB showed anti-predatory behavior throughout the day (in particular, alert and fear) but almost never aggression [45]. Similarly, analyzing the correlation between visitors in a park and animal behavior, Blanchett et al. noted how, as the number of guests and, therefore, the noise levels increased, the birds showed some changes in flight (such as moving away from the visitor pathway, decreased length in their paths and increased use of vegetation cover) but no signs of stress (such as aggression, pacing or feeding, resting, nesting alterations) [46]. However, animal reactions also depend on environmental characteristics and on the type and intensity of the noise. Birds in inhabited areas vocalize at lower frequencies than those in more open areas because lower-frequency vocalizations are more susceptible to energetic masking from anthropogenic noise. This can negatively influence female sexual receptivity and maternal investment in clutch size, as described by Masayuki Senzaki et al. [47]. Elevated noise in excess of 75 dB also tended to increase the frequency of flight [48] and the probability of clutch failure [49]. Amorim et al. have shown that Lusitanian toadfish breeding males exposed to elevated environmental noise produced more live eggs than males exposed to noise generated by boats [49].

However, in the marine environment, the situation is somewhat more complex. Foraging and movements appear to decrease and increase according to individual characteristics (sex, age, body condition), to the environmental context (e.g., food availability) but mostly to the responsiveness to predator sounds [50]; Marine fauna showed changes in behaviors such as anti-predator response with higher swimming speed, school polarization, lower dives, moving away from the noise source [51,52]. Pirotta et al. showed how vessel noise could have an effect on Blainville’s beaked whale (Mesoplodon densirostris) behavior even at relatively large distances, up to 27 km, during the foraging, with a restriction in the movement of groups, a period of more directional travel, a reduction in the number of individuals clicking within the group or a response to changes in prey movement [53]. During sonar exposure, movement was faster and more directed, with displacements away from sonar-producing ships and prolonged intense (<10 bpm) bradycardia [54]. Animals were also more likely to cease calling during exposure [55], to reduce the whistle rates [56], or they exhibit stronger dark avoidance and scotophobia [57].

Another often investigated aspect is the hyperactivity and stress that occurs after intense acoustic exposure. Manukyan et al. showed that mice exposed to noise over 90 dB for 60 days show higher levels of anxiety, alterations in orientation (minimum number of alternation sequences, shorter distance during alternation sequences and more time to complete the sequences of alternation) in addition to an increased cardiovascular risk with an increase in LDL and a decrease in HDL when compared to mice not exposed to noise or to mice who had previously been treated with alpha-blockers [58]. 

Increased activity at the level of the inferior colliculus was noted in pigs 4–12 h after the administration of a strong acoustic stimulus (124 dB) in the cochlea [59]. This was also reported in mice exposed to over 110 dB for one hour [60], resulting in hyperactivity and agitation. Anxiety and agitation in animals caused by loud and sudden noise stimuli such as thunder or fireworks were both reduced when pheromone collars were used, thereby possibly confirming the hypothesis regarding the role of some important structures of the central nervous system [61,62].

Disturbing acoustic stimuli can affect cognitive abilities. Treated mice had a moderate deficit in spatial memory (for example, prolonged time of reaction, increase in wrong entries, a longer escape latency during training sessions and a shorter latency during probe trial sessions) [63]. Similarly, when using the Morris water maze (one of the most widely used tasks in behavioral neuroscience, which provides that testing animals are placed in a large circular pool of water and required to escape from water only using spatial memory) and a series of lever-pressing tasks, it was shown that noise exposure impaired spatial learning, reference memory and stimulus–response habit learning, whereas cognitive flexibility tasks and reversal learning were unaffected [64].

## 4. Discussion

The aim of this study was to analyze the most recent scientific literature so as to try to identify any correlation between exposure to noise sources and the development of neurobehavioral disorders in all animal species. 

The data showed that in the past two years, there was an increase in the number of papers published on the subject, which goes to show that there is an increased awareness and interest regarding environmental issues.

Difficulties, however, remain in studying this complex issue, especially due to the complexity of specifically testing animals and in understanding or measuring their reactions. The fact that the number of publications has increased in the last years reflects the fact that there is an ever-increasing awareness regarding the possible effects that pollutants have on our biodiversity. Most of the studies published are observational studies, probably due to the fact that testing animals is a difficult procedure. The quality of the reviews that were included in this paper had a fair methodology, but the same could not be said of the experimental studies reviewed, as these quite often lacked accurate information regarding the sample that was examined.

Practically all the papers reviewed agree in highlighting a negative deleterious effect due to noise exposure, whether it be acute or chronic, anthropogenic or not.

In particular, within the behavioral sphere, substantially, there were three aspects that came to light, namely, alterations in movements, in foraging and in reproduction. This result is in line with what has been reported in the history of scientific literature, especially in the last decade. Thus, birds have been shown to move to places with less vehicular traffic, as the noise generated by traffic makes it more difficult to detect predators and conceals singing [65]. Birds also tend to emit briefer, more frequent sounds so as to reduce acoustic masking [66,67]. Their ability to predict the aggressive intent of other birds is also influenced by noise-generating sources [68]. Likewise, among marine mammals, changes in vocalization, stress, changes in breathing, increased swimming speed, orientation away from the sound source, sudden and longer-lasting dives, alterations in migratory routes, stranding, and changes for foraging and breeding are detected [69]. For fish and invertebrates, chronic exposure can lead to an alteration of growth and reproductive processes, stress, increased heart rate, increased motility, and alteration of migratory phenomena [70]. 

Another aspect highlighted through our analysis was that the marine ecosystem was the system mostly investigated. This was predictable since the International Community has for years been made aware of the severity of the consequences produced by anthropogenic noise on marine species. In the 1982 Convention on the Law of the Sea, anthropogenic sounds are defined as *‘the direct or indirect introduction, by man, of substances or energy into the marine environment including estuaries, which causes or is likely to cause deleterious effects such as damage to biological resources and marine life (omissis)’* (art. 1) [71]. This was further emphasized by the European Community in the Marine Environment Framework Directive (2008/56/EC), which expressly included, among the forms of pollution, underwater noise, defined as “*the introduction intentional or accidental acoustic energy in the water column, from point or diffuse sources*” [72]. This global issue has recently also come under the scrutiny of numerous international organizations, such as the United Nations (ONU), the Antarctic Treaty Organization (ATS) and the International Whaling Commission (IWC).

However, it is necessary to take into account the various variables that are related to sound. These may interact with each other and may or may not influence the effects on animal species. This review has indicated that there are many factors that need to be considered. The effect that noise has on animals varies greatly between individuals of the same species and between different species due to several factors, which include, amongst others, age, sex, individual sensitivity and previous exposure, with the latter also depending on the characteristics of the noise source, such as its intensity, duration, frequency and type [73]. Therefore, due to these reasons, it is difficult to establish a definitely dangerous noise level, keeping in mind that some effects have already been documented at low noise levels between 40 and 50 dB (A) [20]. One has to remember that almost all terrestrial environments are not inherently silent; there are noises of a geophysical nature (such as rain, waves, ice movements and earthquakes), others of a biological nature (such as those produced by numerous animal species). Furthermore, human activities, such as navigation and transport in general, extraction of gas and oil from the seabed, the search for related deposits, and, above all, the use of active sonar and pinger (often correlated with the spillages and death of cetaceans) all increase marine sound pollution [74,75]. Studies in this area are increasing our knowledge, and some researchers have pointed out that the level of acidification of seas, due to higher and higher quantities of carbon dioxide dissolved in water, tends to reduce the capacity of water itself to adsorb low-frequency sounds. This phenomenon can therefore cause a further increase in underwater noise pollution [76].

Assuming that noise can interfere with the habits and behaviors of animal species, many researchers have started to try to better understand the neurological basis of this evidence. As the theories related to behavioral alterations in humans and as highlighted through this analysis, the most accredited hypotheses involve some areas of the central nervous system and modifications related to neurotransmitters and hormones, such as noradrenergic neurotransmission and adrenocortical activation [77]. For example, scientific evidence has shown that hippocampal theta rhythms in behavioral rats are associated with increased locomotion, agitated behaviors and behavioral arousal in free-moving [78,79]. Similarly, if exposed to airborne noise levels > 80 dB, mice showed higher serum norepinephrine (NE) levels and signs of agitation, such as increased frequency of grooming and defecation, compared to control animals [80]. In fact, NE from the locus coeruleus plays an important role in stress responsiveness, arousal, learning, memory, attention, response to novelty and synaptic plasticity [81,82,83,84,85,86,87,88]. Furthermore, other experiments have highlighted how exposure to intense noise can initiate a cascade of neuroendocrine events similar to the response to stressful events, including activation of the HPA axis [89].

However, the discoveries are constantly evolving as the factors involved are many, and some data are still conflicting. In fact, recent studies conducted on rats have shown how some particular anthropogenic noises can negatively influence neurobehavioral activities only if considered individually. For example, railway noise detects the level of phosphorylated-Ca(2+)/calmodulin-dependent protein kinase II (p-CaMKII), respectively, in the hippocampus, temporal lobe and amygdala, resulting in less synaptic transmission and less learning [90]. However, for the same frequencies, the combined train–aircraft noise had less effect. Animals did not show anxiety, monoamine levels were not increased [91], and there were no alterations in the expressions of phosphorylated calcium/calmodulin-dependent protein kinase II (p-CaMKII) and N-methyl-D-aspartate receptor 1 (NMDAR1 or NR1) in the hippocampus, probably due to smaller intermittency of CTN, CTN’s smaller R-weighted sound pressure level based on rats’ auditory sensitivity [92]. Nevertheless, researchers are continuing to investigate the mechanisms of the noisy sources ’synergy, as the results are not so obvious. In fact, for example, different consequences have been highlighted for short and long-term exposures, respectively, with excitatory or inhibitory effects on the nervous system [93].

It is worth noting that these hormonal and behavioral expressions caused by various stressors (sensory stimuli, drugs, dietary components and psychological stressors) are attributable to genetic modifications that are still to be investigated but attributable to the expression of miRNAs through the modulation of various signaling pathways in the nervous system which responsible for perception from the external environment [94]. In this respect, some authors have observed a significant increase in Taok1 (Tao kinase 1) levels in cases of noise exposure, which in turn is associated with an increase in apoptosis mediated through MAP kinase activation [95,96].

A more recent field of research concerns the changes induced by various kinds of trauma, including exposure to noise, on the synaptic plasticity of the cerebral cortex. In mammals, hearing loss can lead to a reorganization of the auditory cortex (AI) tonotopic map [97]. Some researchers have highlighted a profound reorganization of this map in AI caused by acoustic trauma of about 30 dB and frequencies above 10 kHz in young cats [98]. It may be that the suppression of the cortical response and the possible reorganization of the tonotopic map represent the neurophysiological basis of problems, such as tinnitus and poor speech intelligibility, even in exposed humans [99,100].

One of the strong points of this review is that it is one of a kind, as there does not seem to be another systematic review that addresses this issue in a similar manner, mainly through investigating all animal species. Having had such a review would have given the authors the opportunity to be able to compare findings with possibly being able to highlight differences or changes. 

The limitations of this review stem from the fact that there is a wide range of animal species described in the various papers reviewed. The variation is not only in terms of the quantity of the groups and species but also from the wide-ranging neurobehavioral alterations studied, which at times were presented in a non-standardized unvalidated methodology. Finally, for its purpose of a general overview of disorders, this review does not deeply address habitat, taxon-specific patterns or neurobiological mechanisms.

## 5. Conclusions

The necessity of preserving and protecting the terrestrial environment and the various animal species that inhabit such an environment is becoming more evident. One should aim at trying wherever possible to try to attain synergy between the technological development brought about by man and the well-being of ecosystems. In the near future, it is highly desirable that new intervention opportunities are put in place. These should include increased public awareness, education, government intervention, promotion of government laws and funding to enforce higher safety measures, such as monitoring underwater noise levels, building harmful anthropogenic noise databases and encouraging “areas protected from noise” and “quiet areas”, these could provide natural corridors without any disturbing noises sources, vehicular traffic or interference in acoustics, in the interest of the well-being and safety of the fauna [101].

## Figures and Tables

**Figure 1 ijerph-20-00591-f001:**
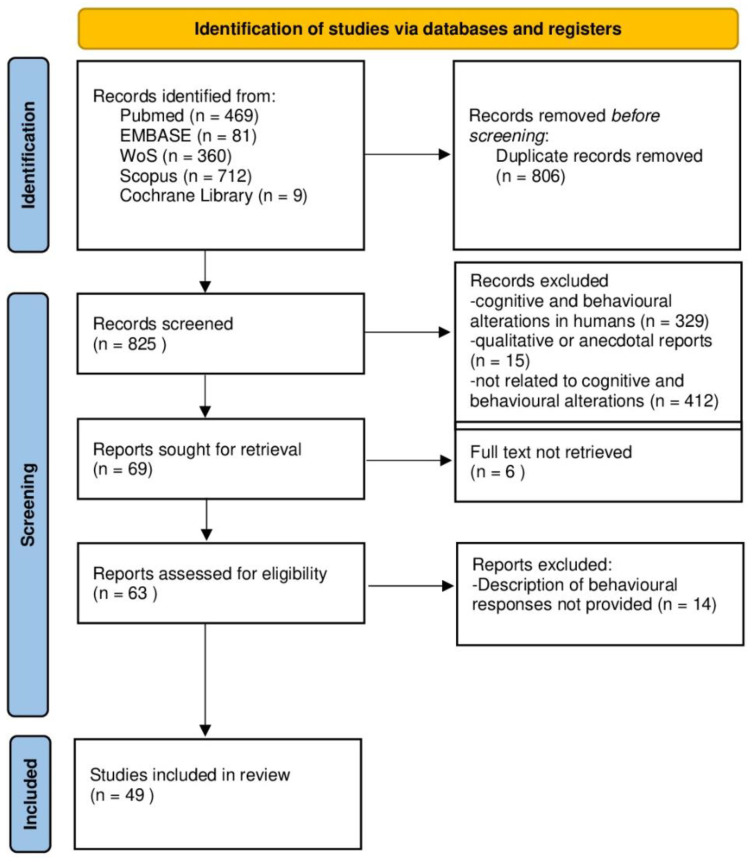
Flow Chart of the bibliographic search.

**Table 1 ijerph-20-00591-t001:** PICO strategy.

Population	Animal with No Differences in Species, Habitat or Geolocation.
Interventions	Noise exposure from any source
Comparison	N.A.
Outcomes	Neurobehavioral or neurological alterations

**Table 2 ijerph-20-00591-t002:** Included studies in this systematic review, in alphabetical order, with their relative score.

First Author.	Year	Country	Type of Study	Species	Alterations	Score
Abdullah	2020	Indonesia	observational study	Elephants	altered anti-predatory’reaction	N.6
Akefe	2020	Nigeria	experimental study	Rats	learning, short-term memory, sensorimotor reflex	J.2
Amorim	2022	Portugal	Case-control	Fish	Behavioral and reproductive responses	N.5
Baltzer	2020	Wadden sea	experimental study	marine mammals, fishes	altered movements, swimming speed, anti-predatory reaction	n.a.
Blanchett	2020	USA	observational study	Birds	aggression, pacing, nesting etc	N.6
Codocedo	2016	Australia/Chile	narrative review	mice, rats	anhedonia, anxiety, social-avoidance behaviors	I.6
Cox	2017	Canada	meta-analysis	Fishes	complex movements and swimming abilities	A.6
Criddle	2018	USA	experimental study	Hamsters	hyperactivity	J.2
De Soto	2016	Spain	narrative review	marine invertebrates	altered movements, swimming speed, metabolic parameters	I.5
Di Franco	2020	Italy/France	systematic review	marine invertebrates, fishes	altered movements, swimming speed, anti-predatory reaction	A.4
Durbach	2021	UK	Observational study	Whales	Behavior responses	N.6
Frouin-Mouy	2020	Mexico	experimental study	Whales	resting, interaction mother-calf	n.a.
Gang	2021	China	Case-control	Mice	Stress response, cognitive capacities, neuroinflammation	N.5
Grunst	2021	Belgium	Experimental study	birds	Parental behaviors	J.3
Hastie	2021	UK	Experimental study	Grey seals	Foraging behavior	n.a.
Heinrichs	2010	USA	narrative review	Rodents	anxiety, hyperactivity	I.5
Hubert	2020	North sea	experimental study	Fishes	changed swimming	n.a.
Issad	2021	Algeria	Experimental study	Gerbils	Circadian rhythm and anxious behavior	n.a.
Kight	2011	USA	narrative review	rats, zebra	cognition, sleep	I.5
Koorpivaara	2017	USA	experimental study	Dogs	anxiety, fear	J.4
Kunc	2016	Uk	narrative review	marine species	aggression, hunting, movements, anti-predatory reaction	I.5
Landsberg	2015	Canada	case-control	Dogs	anxiety, fear	N.6
Lara	2021	China	Case-control	Larval zebrafish	stronger dark avoidance, scotophobia, movements and swimming alterations	N.5
Leduc	2021	Brazil	Experimental study	Fish	Cognitive performance	J.2
Li	2018	Indo-Pacific sea	narrative review	Dolphins	altered movements and vocals	I.4
Longenecker	2016	USA	cohort study	Mice	hyperactivity	N.6
Mandel	2016	Israel, Uk	narrative review	cows, calves	various	I.4
Manukyan	2020	Armenia	case-control	Rats	anxiety, memory	N.6
Martin	2022	France	Experimental study	Cape fur seals	Behavioral responses	J.2
Mikolajczak	2013	Poland	experimental study	Geese	movements, stress	J.2
Miller	2022	UK	Observational study	Cetaceans	Foraging behavior	N.a.
Mills	2020	Polynesia	experimental study	Fishes	hiding, distance, aggression	J.2
Mulders	2013	Australia	case-control	Pig	hyperactivity	N.5
Nabi	2018	China/USA	narrative review	marine mammals	masking, altered reproduction	I.5
Park	2022	Korea	Case-control	Frogs	behavioral–physiological–immunological response	N.6
Pellegrini	2020	Brazil	Observational study	Dolphins	Foraging behavior	N.6
Peng	2015	China	narrative review	marine species	nesting, aggression, anti-predatory reaction	I.6
Pienkowski	2011	Canada	narrative review	rats, cats	cortical plasticity	I.4
Pirotta	2012	USA	case-control	Whales	foraging, movements	N.6
Popper	2019	USA/Uk	narrative review	Fishes	impairment of spawning, interference with foraging, disruption in migration-habitat selection	I.4
Samson	2016	USA/Netherland	narrative review	cephalopods	escape, inking, altered speed	I.3
Senzaki	2020	USA	Observational study	Birds	Reproductive behaviors	N.6
Shannon	2016	USA	systematic review	Wildlife	vocals, movements, foraging, escape, vigilance, mating	A.5
Uran	2012	Argentina	experimental study	Rats	recognition, memory	J.2
Van der Knapp	2021	Netherlands	Observational study	Fish	Behavioral responses	N.5
Van der knapp(b)	2021	Netherlands	Observational study	Fish	Movement behavior	N.6
Wang	2022	China	Observational study	waterbirds	Flight pattern	N.6
Wieczerzak	2021	Canada	Cohort study	Mice	Cognitive behavior	N.6
Williams	2022	USA	Case-control	narwhales	Locomotor reactions	N.6

N.a. = not applied; N = New Castle Ottawa Scale, J = Jadad scale, A = AMSTAR scale.

**Table 3 ijerph-20-00591-t003:** Reviews included in the study.

First Author	Included Articles	Level’ Noise	Results
Codocedo	narrative	not specified	In rats, noise exposure for 24 h generates a decrease in several miRNAs, including miR-183, leading to adecrease in the level of the target TaoK1, which participates in the activation of the MAPK pathway and the induction of cell apoptosis
Cox	42	not specified	Increased hearing thresholds and cortisol levels were associated with an increase in stress-related hormones, and suggest that anthropogenic noise has the potential to cause both short- and long-term physiological effects
De Soto	15	157/136–162/156–168 dB re 1 μPa	noise interferes with growth larvae, metabolism, reproductive rates, changes in swimming and movements
Di franco	57	not specified	acute and chronic marine noise can cause a wide variety of effects on marine invertebrates and vertebrates, such as swimming and gregarious patterns, anti-predator responses, mating and spawning patterns, auditory damage, communication masking, changes in habitat use, migration and displacement, stress-related physiological responses
Heinrichs	narrative	120 dB–12 kHz	mechanisms that can induce hyperactivity in animals exposed to stressors, such as loud noises, are related to hippocampal changes, in the locus coerulus or to activation of adrenocortical hormones.
Kight	narrative	65–95–110 dB	noise stressed animals are not able to reproduce species-appropriate vocalisations, they do spatial errors and stress during pregnancy but noise might act as a beneficialstimulant of brain activity, such as white noise during sleep
Kunc	narrative	not specified	Noise may also negatively affect the social structure between pairs and groups, can impede defence against predators, reduce the ability to maintain territories or alter the reproductive behavior
Li	narrative	pulse with sound exposure levels (SELs) > 183 dB re: 1 μPa2 and nonpulses > 195 dB re: 1 μPa2s	dolphins with vessel noise change their fluke, rate, heading, dive depth and reduced their sounds
Mandel	narrative	not specified	in cows white noise or classical music decreases stress level
Nabi	narrative	not specified	masking can compromise reproduction, mother-offspring bonding, foraging and survival because animals are unable to interpret and respond to mating calls, offspring calls, prey sounds or predator sound
Peng	narrative	119–250 dB re 1 μPa	the effects of anthropogenic noise on marine organisms are dependent on the species investigated and both the levels of impulsive and stationary noise
Pienkowski	narrative	68–72 dB spl	sounds can lead to a reorganization of auditory cortex not unlike that following restricted hearing loss but different from that learning-induced
Popper	narrative	20–50 Hz (bulk), 180 to 200 dB re 1 μPa2 s^−1^ (pile drivers), <1 Hz (vessel)	change in behavior from small and short-duration movements to changes in migration routes and leaving a feeding or breeding site; decrease in detectability ofbiologically relevant sounds (e.g., sounds of predators and prey, sounds of conspecifics, acoustic cues used for orientation)
Samson	narrative	20–1000 Hz	in cephalopods, reactions considered to be escape and/or startle behavior (blanching, jetting, inking) mostly occurred at low frequencies and high sound levels
Shannon	188	52 and 68 dBA SPL re 20 μPa (terrestrial)/67–195 dB SPL re 1 μPa (acquatic)	noise cause increased stress levels, decreased reproductive efficiency, impacted the vocal behavior and reduced the foraging efficiency

**Table 4 ijerph-20-00591-t004:** Cohort, case-control, experimental and observational studies included in this review.

First Author	Sample	Level of Noise	Aim	Lenght of Study	Results
Abdullah	2	20–75 dB	exposure to various noise for 15 min	3 repetitions in each day for 5 days	noise interferes with prey perceptions of predators
Akefe	30	100 dB	exposed to noise, with or not kaempferol + zinc gluconate	48 days	noise interferes with oxidative stress
Amorim	16	104–140 dB re. 1 μPa	impacts of boat noise exposure in the reproductive success of wild toadfish	2 weeks	Noise affected reproductive success by decreasing the likelihood of receiving eggs, the number of live eggs and increasing the number of dead eggs
Baltzer	not specified	120–99 dB re 1 μPa2s	effects of underwater noise on marine mammals	1 day	anchor pipe vibration embedment noise might induce a behavioral reaction (changes in movements)
Blanchett	98	51.5–66.6 dB	associations between visitor numbers, noise levels and stress or critical behavior	12 days	lack of association between visitor numbers and stress or critical behavior
Criddle	24	85–115 dB	NMDA receptor blocker and sound exposure	4 h + 28 days	treated animals show lower hyperactivity
Durbach	Not specified	approximately 3 kHz and a nominal source level of 235 dB re 1 µPa	investigate the effect of sonar activity on movement behaviors	3–4 days for 3 years	Faster and more directed movement during sonar exposure; animals were more likely to cease calling during exposure
Frouin-Mouy	2	94.8–110.2 dB re 1 μPa	measuring the underwater source levels, behavioral vocal and non-vocal marine mammal signals	1 month	noise can interfere communications between group
Gang	120	mean sound pressure level of 72 dB (A)	Associations between aircraft noise and cognitive functions	2 h daily for 4 days	Changes in spatial recognition memory
Grunst	34 pairs	60 dB	altered parental behaviors in response to consistent freeway noise and a diverse anthropogenic noise	2 weeks	no population-level changes in nestling provisioning behavior during noise but individual differences in noise sensitivity
Hastie	5	148 dB re 1 µPa	measuring the relative influence of a sound (silence, pile driving, and a tidal turbine) on decision-making and foraging success in grey seals	8 days	Foraging success was significantly reduced (16%–28% lower) when the speaker was located at the Low Density prey patch
Hubert	64	mean SPLs 128.3 or 119.0 dB re 1 μPa	exposed seabass to different impulsive sound treatments (pulse level, elevated background level)	3 sound treatment in each day for 2 days	upon sound exposure, fishes increased their swimming depth
Issad	32	80 dB	Effects of light and noise pollution on body temperature and anxious behavior	3–4 weeks	significant decrease in the number of line crossings and time spent in the open field test.
Koorpivaara	182	not specified	dexmedetomidine for noise-associated acute anxiety and fear in dogs	3 months	noise can caused hyperactivity by locus coerulus’activation
Landsberg	24	average 83.9 dB	two treatment groups (DAP and placebo) in response to a thunderstorm recording	a week	pheromones reduce anxiety and fear by noise
Lara	Not specified	130 and 150 dB re 1 μPa	Shipping activity can altered fish’behavior	5 days	continuous noise can increase dark avoidance in anxiety-related dark/light preference test and impaired spontaneous alternation behavior
Leduc	32	45–100 dB	Noise can reduce the available cognitive processing capacity	3 weeks	fish exposed to noise playbacks require additional time to reach this target and reduce exploratory behavior
Longenecker	16	116 dB	relationship between tinnitus, hearing loss, hyperactivity and bursting activity post noise trauma	1 h	noise increase tinnitus and hyperactivity
Manukyan	24	91 dB	monitoring hdl, ldl, cholesterol, cognitive functions post noise exposure	60 days	chronic noise altered behavioral activity, delay in movement and orientation, increased anxiety, deficit spatial memory
Martin	35 groups (369 individuals)	60.9–64.4 (low)/64.4–70.5 (medium)/70.5–80 dB re 20 μPa RMS SPL (high)	Effect of car and boat noise on marine mammals behavior	1 month	detriment of vital activities such as resting and nursing that decreased considerably (from 5.9 to 45% decrease)
Mikolajczak	40	94–104 dB	effect of noise by wind turbines on the stress parameters (cortisol)	17 weeks	lower activity, some disturbing changes in behavior, increased cortisol
Miller	43	1–4 kHz	Effect of sonar noise on foraging	13 h	whales ceased foraging completely during killer whale and sonar exposures
Mills	28/20	120–70 dB re 1 μPa2s/Hz	short-longer effect motorboat-noise playback on the behavior, cortisol, androgens of anemonefish	30 min/48 h	in short term, hiding aggression, androgen level increased
Mulders	24	20–120 dB	monitoring hyperactivity post noise exposure	2 weeks	hyperactivity in the colliculus begins at some time between 4 and 12 h post trauma
Park	27/24	41.3–57.60 dB	Effects of wind turbine on frog’s behavior	2 days	Call rate increased after 1 h of exposure
Pellegrini	122 groups	180 dB re: 1 lPa V^−1^	Effects of boat noise on foraging	9 months	cooperative foraging may potentially be reducedor interrupted by the presence of boats, in response to the number, type and speed, indicating a behavioral change and acoustic masking
Pirotta	32	50–200 dB re 1 μPa2s/Hz	How vessel noise influenced foraging behavior	5 days	ship noise caused a significant change in whale behavior up to at least 5.2 km away from the vessel.
Senzaki	142 species (58,506 nest)	Not specified	Effect of light and noise on reproductive success	14 years	Closed-habitat, but not open-habitat, birds also tended to experience a decline in clutch size with noise exposure
Uran	30	95–97 dB SPL	Monitoring hippocampal-related behavioral alterations	15–30–45 postnatal day	moderate intensity can changed hippocampus, with observed behavioral effects
Van der Knapp	49–250	123–140 dB (re 1 μPa)	Effect of boat noise on behavior	6 months	in presence of boat noise) fishes spent more time in behaviorsconsidered to be a response to predators
Van der knapp(b)	14	114–138 dB (re 1 μPa)	Effect of wind turbine on movement behavior of free swimming	4 months	cod did not increase their net movementactivity, but moved closer to the scour-bed (i.e., hard substrate), surrounding their nearest turbine,
Wang	60	60–100 dB	investigate the effects of ship noise on foraging, vigilance and flight behaviors	1 month	As the noise level increased, foraging behavior decreased and vigilance and flight behaviors increased, particularly above 70 dB
Wieczerzak	10/11	40 Hz	Investigated neural plasticity in the auditory and prefrontal cortices in the days following noise exposure	1 month	noise exposure impaired spatial learning and reference memory
Williams	13	241 dB re 1 μPa-m	Investigated reactions to anthropogenic noise by this deep-diving cetacean	5 years	movement from surface to depth (descent) was often more gradual for control dives than for noise exposed dives which showed shorter, more rapid ‘directed’ descents

## Data Availability

Not applicable.

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
