# Peer review of "Neurobehavioral Alterations from Noise Exposure in Animals: A Systematic Review"

_ijerph, 2022, doi:10.3390/ijerph20010591_

Round 1
Reviewer 1 Report
After reviews by Kight & Swaddle 2011, Kunc et al. 2016; and Shannon et al. 2016, this is another general review about impact of anthropogenic noise across taxa. The authors chose to cover the last 12 years from 2010-2022. The title suggests a more neurobiological focus than previous reviews, but I did not find that back in the main text. I believe this is an interesting and worthwhile review. I personally do not like the extensive search report explanations and tables, which I do not believe add much to the story in this particular paper. The review is an interesting compilation of studies, but is limited in addressing habitat or taxon-specific patterns and, as mentioned, does not zoom in explicitly into the role for or impact on neurobiological processes, which would certainly give added value.
Author Response
Dear Reviewer 1,
We would like to thank you for your valuable observations about our work. We addressed your comments and suggestion below (in italics).
- After reviews by Kight & Swaddle 2011, Kunc et al. 2016; and Shannon et al. 2016, this is another general review about impact of anthropogenic noise across taxa. The authors chose to cover the last 12 years from 2010-2022. The title suggests a more neurobiological focus than previous reviews, but I did not find that back in the main text.
Thank you for this observation, which we can’t help but agree. The aim of this review does not aspire to explain in depth biological patterns and neurophysiological mechanisms. Instead, it aims to provide a focus on the detrimental neurobehavioral effects that noise can have on animals of all species, as well as it has on humans. In addressing this issue, it was necessary also to provide some insights on biological mechanisms, but it was not our first aim. We recognize this issue in our work, and the fact that this was not properly addressed in Introduction. Therefore, we provided some clarifications both in Introduction, so that readers would not be misled (line 84)
- I believe this is an interesting and worthwhile review.
Thank you very much for your consideration.
- I personally do not like the extensive search report explanations and tables, which I do not believe add much to the story in this particular paper.
The in-depth description of methods is required by PRISMA guidelines, the methods we followed to conduct this systematic review. However, we tend to agree with your comment, therefore we moved Table 1 in Supplementary Material. We hope that now the paragraph would be more readable. Thank you for your comment.
- The review is an interesting compilation of studies, but is limited in addressing habitat or taxon-specific patterns and, as mentioned, does not zoom in explicitly into the role for or impact on neurobiological processes, which would certainly give added value.
As mentioned before, we tried to clarify this issue in Introduction, and this specific limitation in Discussion (line 477).
Thank you again for the time and effort spent in reading and reviewing our manuscript.
Sincerely,
The Authors

Reviewer 2 Report
Congratulations for this review. Papers like yours are necessary in a changing world.
Some comments and considerations:
Please, consider a more in-depth English review. Especially with regard to connectors and structuring of sentence elements.
I suggest exploring the first paragraph more by citing works regarding the growing concern about the increase of noise levels both on land and in the oceans.
Table 1: the word “child” was used on the search or it was a result of a translation?
The the meaning of the abbreviations can be placed at the bottom of the table
Line
90: could you explain better the type of noise exposure (when you detail pollution caused by human or not)?
92: It would be clearer if the short term effects examples were given separated from the other (medium-long term effects).
112 and 114: I suggest to insert the meaning (when possible) of acronyms such as AMSTAR and JADAD and others that appear throughout the text.
Last line from Table 3: narwhals instead of “narwhales”
207: Please pay attention in the translation. Although being capable of inking, the literature cited did not mention this behavior as a response to sound stimuli in octopuses. Maybe you meant cuttlefish (sepias)?
245: It is not clear if the animals increased/decreased vocalization rates or if it was about the presence/absence of vocalization behavior.
252: I suggest “sexes” instead of genres
270: What is UV acoustic signal?
273: Female from which taxa?
308: Please, specify that the paper cited is about toothed whales or even beaked whales (only toothed whales are capable of the clicking behavior referred).
314: It is not clear if the animals were more likely to increase or reduce whistles or if it was about the presence/absence of whistling behavior.
333: An explanation of what is the Morris Water Maze mentioned (even if short and between parenthesis) is very welcome.
359: I advice to include a word like “tend” on the sentence: They tend to emit shorter sounds.
397: Reorganize the sentence and choose/include different words to avoid redundancy.

Author Response
Dear Reviewer 2,
we would like to thank you for the time and effort spent in reviewing our paper. We addressed your valuable comments below and you can find the changes in the manuscript.
- Congratulations for this review. Papers like yours are necessary in a changing world.
Thank you for your consideration.
- Please, consider a more in-depth English review. Especially with regard to connectors and structuring of sentence elements.
The paper was thoroughly reviewed for the English style by a native English speaker, thank you.
- I suggest exploring the first paragraph more by citing works regarding the growing concern about the increase of noise levels both on land and in the oceans.
Thank, you we added some references to provide to depict the scenario more deeply (from line 37 to 49).
- Table 1: the word “child” was used on the search or it was a result of a translation?
Thank you for your comment. Firstly, following the suggestion of another reviewer we decided to move Table 1 in supplementary material. You can find the world “child” in the terms of search because we used the Embase tool which automatically includes in the search all he synonyms of the world you entered. So, we entered ‘'behavior disorder' and in Embase synonyms include terms related to children disturbs. This is a way to broad the online search, and of course during the selection, as explained, we excluded all the studies related to humans.
- The the meaning of the abbreviations can be placed at the bottom of the table
We added a footnote with the meaning of the abbreviations.
- 90: could you explain better the type of noise exposure (when you detail pollution caused by human or not)?
We clarified, thank you
- 92: It would be clearer if the short term effects examples were given separated from the other (medium-long term effects).
Thank you for this comment. In summarizing reviews this would be very hard to accomplish, while you can see in Table 4 (previously Table 5) that the time of observation was indicated (length of study) and this provides information about the timing in which the effects were seen. We clarified it in line 187.
- 112 and 114: I suggest to insert the meaning (when possible) of acronyms such as AMSTAR and JADAD and others that appear throughout the text.
We added them, thanks, for AMSTAR and PRISMA. JADAD is not an acronym but the name of the inventor of the method. Also we added footnotes for table 3.
- Last line from Table 3: narwhals instead of “narwhales”
Thank you we corrected
- 207: Please pay attention in the translation. Although being capable of inking, the literature cited did not mention this behavior as a response to sound stimuli in octopuses. Maybe you meant cuttlefish (sepias)?
Thank you, we amended the sentences, we intended, as shown in the reference, cephalopods like S. Officianilis, so cuttlefishes.
- 245: It is not clear if the animals increased/decreased vocalization rates or if it was about the presence/absence of vocalization behavior.
In the study, the owners of the dog rated some behaviors including vocalizing on a scale from 0 to 4. It was demonstrated an effect of the drug which acts on anxiety patterns in reducing this anxiety-related behaviors including vocalizing. This clarification was added in the main text.
- 252: I suggest “sexes” instead of genres
Thank you, we corrected.
- 270: What is UV acoustic signal?
That was a spelling mistake. It is UAV, Unmanned Aerial Vehicle, a drone technology. We clarified it in the text
- 273: Female from which taxa?
We specified, thanks.
- 308: Please, specify that the paper cited is about toothed whales or even beaked whales (only toothed whales are capable of the clicking behavior referred).
The study was specifically about Blainville’s beaked whale (Mesoplodon densirostris), we specified in the text, thank you.
- 314: It is not clear if the animals were more likely to increase or reduce whistles or if it was about the presence/absence of whistling behavior.
We specified thank you.
- 333: An explanation of what is the Morris Water Maze mentioned (even if short and between parenthesis) is very welcome.
We provide a brief explanation.
- 359: I advice to include a word like “tend” on the sentence: They tend to emit shorter sounds.
Thank you, we corrected.
- 397: Reorganize the sentence and choose/include different words to avoid redundancy.
Thank you, we rephrased it.
Again, we thank Reviewer 2 for his/her valuable comments which helped in improving our manuscript.
Sincerely,
The Authors
